# Seed Germination within Genus *Rosa*: The Complexity of the Process and Influencing Factors

Roxana L. Stoian-Dod [1], Catalina Dan [1,*], Irina M. Morar [2], Adriana F. Sestras [2], Alina M. Truta [2], Gabriela Roman [1,3] and Radu E. Sestras [1]

1    Faculty of Horticulture and Business in Rural Development, University of Agricultural Sciences and Veterinary Medicine, 3-5 Mănăștur Street, 400372 Cluj-Napoca, Romania; roxana.stoian@usamvcluj.ro (R.L.S.-D.); gabrielaroman33@yahoo.com (G.R.); rsestras@usamvcluj.ro (R.E.S.)

2    Faculty of Forestry and Cadastre, University of Agricultural Sciences and Veterinary Medicine, 3-5 Mănăștur Street, 400372 Cluj-Napoca, Romania; irina.todea@usamvcluj.ro (I.M.M.); adriana.sestras@usamvcluj.ro (A.F.S.); alina.truta@usamvcluj.ro (A.M.T.)

3    Horticultural Research Station, 400457 Cluj-Napoca, Romania

*    Correspondence: catalina.dan@usamvcluj.ro

**Abstract:** Seed germination is a crucial stage in the life cycle of plants, and understanding the factors influencing germination is essential for successful cultivation, plant breeding, and conservation efforts. The genus *Rosa*, commonly known as roses, encompasses a diverse group of flowering plants renowned for their beauty and fragrance. *Rosa* germination is influenced by a variety of factors, including seed dormancy, environmental conditions, and seed treatments. Many *Rosa* species exhibit different types of seed dormancy, such as physical dormancy caused by hard seed coats and physiological dormancy due to internal mechanisms. Overcoming seed dormancy often requires specific treatments, including cold stratification, scarification, or chemical treatments, to promote germination. Environmental factors, including temperature, moisture, light, and substrate, play vital roles in *Rosa* germination. Temperatures ranging from 15 to 25 °C, moisture, and exposure to light or darkness, depending on the species, constitute suitable conditions for seed germination. Many studies have been conducted to investigate the germination requirements of different *Rosa* species, thereby expanding our understanding of their propagation and conservation. Additionally, advancements in techniques such as in vitro germination and molecular approaches have further enhanced our understanding of *Rosa* germination biology.

**Keywords:** gibberellic acid; H$_2$SO$_4$; methods; roses; scarification; seed stimulation; stratification

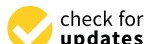



## 1. Introduction

### 1.1. Distribution of the Genus Rosa

With more than 200 species spread across the northern Hemisphere, the *Rosa* genus within the Rosaceae family is one of the most significant ornamental plant genera regarding economic and cultural history [1,2]. The genus is divided into four subgenera according to conventional taxonomy [3], three of which are monotypic: *Platyrhodon* (Hurst) Rehder, *Hulthemia* (Dumort.) *Focke*, and *Hesperhodos* Cockerell. Approximatively 95% of all species are found in the subgenus *Rosa*, which is split into 10 sections, one of which is *Caninae*, the subject of this review. The genus is native to North America, East Asia, and Europe/West Asia (Figure 1).

Roses are among the most significant and versatile horticultural and industrial products. According to Guimares [4], they can be utilized as cut or garden flowers. Additionally, roses have long been employed in the cosmetics, food, and perfume industries [5]. Strlsjö and Larsen [6,7] stated that fruits (rose hips) are a good source of bioactive substances, such as vitamin C, carotenoids, tocopherol, phenolic acid, bioflavonoids, tannin, pectin, organic

acids, amino acids, essential oils, and unsaturated fatty acids. Rose hips are composed of 29% seed and 71% pericarp, and the hues range from red to orange.

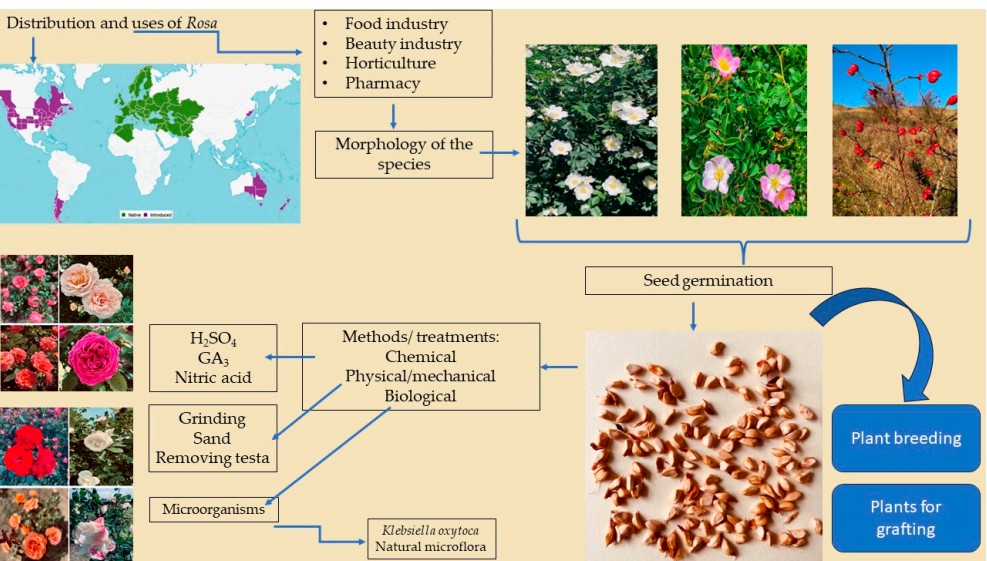

**Figure 1.** Distribution, uses, and seed germination in the *Rosa* genus.

The species reproduces through layering, cutting, tissue culture, basal shoots, and the use of its seeds [8]. Due to its resistance to drought stress, it serves as the most significant stock of ornamental roses [9].

Dog roses, which are part of the section *Caninae* (DC.) Ser. and dominate the territory of Europe and West Asia, are crucial to the development of rootstocks for the breeding of ornamental roses. Due to their distinct meiotic behavior and breeding system, dog roses occupy a special place among plants [9,10].

### 1.2. Botanical Aspect of the Species

The dog rose is a deciduous shrub that typically grows 1 to 5 m tall; however, it can, occasionally, climb higher into the crowns of taller trees by using the little, pointed, hooked prickles that cover its stems. The bark is light brown and very spiny, whereas the plant has spines, prickles, or thorns [5].

The leaves are compound, have 5–7 glabrous leaflets that are pinnate and have serrate margins, and are alternately inserted along the stem. Stipules are present at the base of the petiole. When injured, leaves release a pleasant scent [5].

The dog rose blooms between June and July, producing fragrant, sweet-smelling petals that are often pale pink, but they can also vary from deep pink to white (Figure 2). They have five petals and are 4–6 cm in diameter. It possesses a specific aestivation, typical of roses. There are three or more scales on the winter buds, and they overlap like shingles, with one edge covered and the other edge exposed. Unusually, though, two of its five sepals are whiskered on both sides, two are smooth, and one is hairy (or bearded) on just one side when viewed from beneath [1,2,5].

In general, an oval, red-orange, 1.5–2.0 cm hip develops after fertilization. Rose hip is a pseudo-fruit (botanically known as hypantium), developed from the inferior gynoecium and receptacle of the flower. It has a fleshy consistency, actually changing from green to orange or flaming red as it ripens (Figure 3). In terms of shape, hips may be globular, ellipsoid, obovate, pear-shaped, or bottle- or flask-like, and they vary greatly in size [2]. *Rosa* hips have smooth and polished-looking epicarps that might be glabrous, while others are dull and thorny. Many have resinous hairs, called trichomes. The hips have a sepal-like cap on top with variable aspect, depending mainly on length and margins. As the summer season progresses, a red, fleshy layer called the pericarp develops, and inside the aggregate pseudo-fruit, seeds mature. These seeds represent the real fruit of the species, being known

as achenes, botanically classified as dry fruits, with one proper seed inside. They are 4.5–6.0 mm long, unevenly shaped, hairy, and yellowish in color [5].

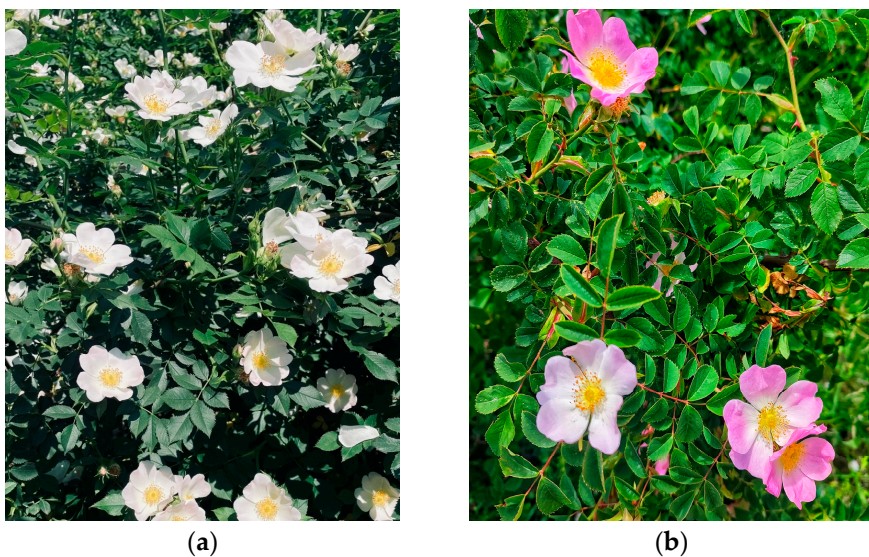

(**a**)  (**b**)

**Figure 2.** Flowers of *R. canina* of (**a**) white color and (**b**) pink hues, from the spontaneous flora of Transylvania, northwestern Romania (photo by R.L.S.-D.).

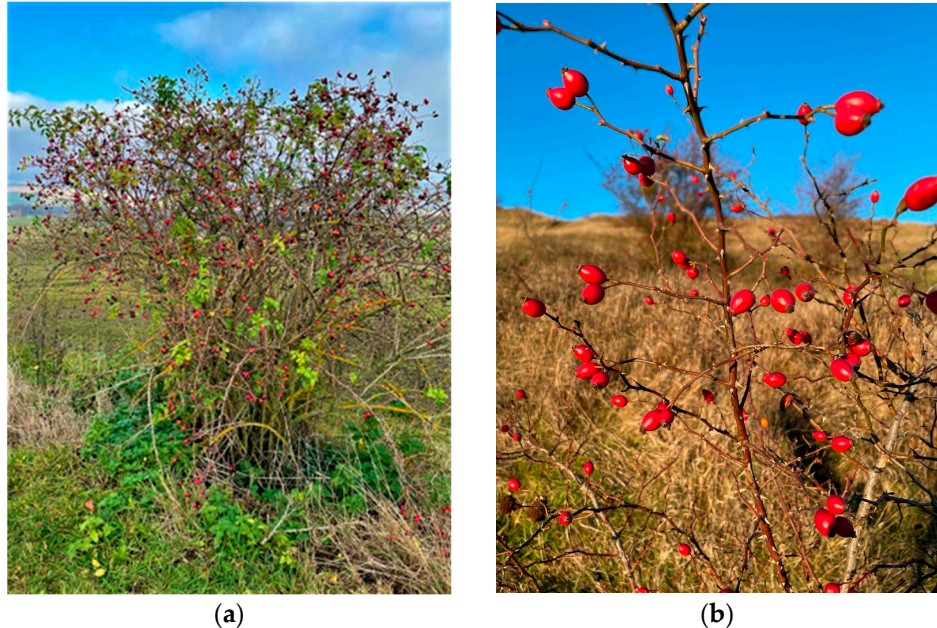

(**a**)  (**b**)

**Figure 3.** (**a**) *Rosa canina* plant (shrub); (**b**) *Rosa canina* pseudo-fruit (hypantium). Bushes photographed in the spontaneous flora of Transylvania, northwestern Romania (photo by R.L.S.-D.).

### *1.3. Importance of the Species*

1.3.1. Chemical Composition of the Hips

The fruits of the rose species are extremely beneficial to human health because of their remarkable quality and quantity of organic and inorganic components. The chemical composition of the hips can be found in Table 1 [11,12].

**Table 1.** Chemical composition of rose hip.

| Nutrient | Value per 100 g | References |
|:---:|:---:|:---:|
| Water | 58.66 g * | [13] |
| Energy | 162 kcal * | [13] |
| Protein | 1.6 g * <br> 0.36 g | [13] <br> [14] |
| Total lipid | 0.34 g * | [13] |
| Ash | 1.18 g * | [13] |
| Carbohydrate | 38.22 g * | [13] |
| Fiber | 24.1 g * | [13] |
| Sugars, total | 2.58 g * <br> 13.28 g | [13] <br> [14] |
| Minerals | | |
| Calcium | 169 mg * <br> 944 mg | [13] <br> [11] |
| Iron | 1.06 mg * <br> 1.1 mg | [13] <br> [11] |
| Magnesium | 69 mg * <br> 122.6 mg | [13] <br> [11] |
| Phosphorus | 61 mg * <br> 122.4 mg | [13] <br> [11] |
| Sodium | 4 mg * <br> 15.8 mg | [13] <br> [11] |
| Zinc | 0.25 mg * <br> 1.3 mg | [13] <br> [11] |
| Copper | 0.113 mg * <br> 0.4 mg | [13] <br> [11] |
| Manganese | 1.02 mg * <br> 5.9 mg | [13] <br> [11] |
| Potassium | 1025 mg | [11] |
| Vitamins | | |
| Vitamin C | 426 mg * <br> 411.0 mg <br> 643.38 mg | [13] <br> [11] <br> [14] |
| Vitamin B-6 | 0.076 mg * | [13] |
| Vitamin A, IU | 4345 IU * | [13] |
| Vitamin E (alpha tocopherol) | 5.84 mg * <br> 34.20 µg/g | [13] <br> [11] |
| Tocopherol beta | 0.05 mg * | [13] |
| Tocopherol gamma | 1.34 mg * | [13] |
| Tocopherol delta | 0.14 mg * | [13] |
| Vitamin K | 25.9 µg * | [13] |
| Carotene alfa | 31 µg * | [13] |
| Carotene beta | 2350 µg * <br> 2.60 µg/g | [13] <br> [11] |
| Cryptoxanthin, beta | 483 µg * | [13] |
| Thiamin | 0.016 mg * | [13] |

**Table 1.** *Cont.*

| Nutrient | Value per 100 g | References |
|---|---|---|
| Riboflavin | 0.166 mg * | [13] |
| Niacin | 1.3 mg * | [13] |
| Lycopene | 6800 µg *<br>390 mg/kg DW | [13]<br>[14] |
| Lutein + Zeaxanthin | 2001 µg * | [13] |
| Pantothenic acid | 0.8 mg * | [13] |

* Nutrient Data Laboratory, ARS, USDA National Food and Nutrient Analysis Program, Wave 9j, 2005, Beltsville MD USA.

### 1.3.2. Health Benefits of Rose Hips

The pseudo-fruit of the rose plant known as rose hip is well-known for being a beneficial source of vitamin C and polyphenols [5]. Studies conducted both in vitro and in vivo have shown fruits' anti-inflammatory, antioxidant, and antiobesogenic properties. The great variety of bioactive substances included in rose hip, particularly the anti-inflammatory galactolipid, have been linked to the health advantages of using the plant [14].

The fact that hips' chemical composition varies based on the genotype, growing area, climate, maturity, cultivation practice, and storage circumstances is an intriguing aspect. A wide range of biochemical activities, such as antioxidant, antimutagenic, and anti-carcinogenic capabilities, are present in phenols. *R. canina* contains 96 mg GAE/g DW of total phenols, according to one study [15]. The high ascorbic acid content of Rosaceae fruits may also play a role in their physiological actions. Numerous biochemical processes, such as antioxidant and anti-carcinogenic characteristics, are carried out by ascorbic acid. Necessary fatty acids, mainly long-chain polyunsaturated fatty acids that humans must consume since our metabolism cannot synthesize them, are another essential component of rose hip. Linoleic and -linolenic acids (ALA) are important fatty acids that control a variety of physiological reactions, including inflammation, immunological function, blood viscosity, and many others [15]. *R. canina* has 1.78% total fat [15]. It contains seven major fatty acids, according to fatty acid analysis: lauric acid (12:0), palmitic acid (16:0), linoleic acid (cis-C18:2 6), -linolenic acid (cis-C18:3 3), nonadecylic acid (19:0), cis-C19:1 6, and cis-C22:2 6 [15]. Another significant component of rose hip that deserves to be mentioned is the galactolipid GOPO. With no known toxicity, this bioactive molecule provides research evidence supporting both anticancer and anti-inflammatory effects [7].

*R. canina* hips appear to have many positive health benefits for humans, but more research is needed to properly understand how they could improve people's wellbeing before they can be regularly advised.

### 1.3.3. Horticultural and Economical Importance of the Species

*Rosa* is one of the most economically important genera in ornamental horticulture. Roses have the largest economic value of all the ornamental plants and have a long and well-documented tradition in selection and breeding [16]. High degrees of heterozygosity, documented issues with sexual reproduction from pollination through seed germination, and various levels of ploidy present challenges for rose breeding [17]. It includes many mountain species, natural hybrids, and cultivars of horticultural interest. It has ornamental value or is used as a soil improver species, and it is used as a rootstock for obtaining many types and cultivars of roses [18].

Roses are propagated both by seed and vegetatively. Seed propagation is practiced in rose breeding when different parental forms are crossed, and the hybrids obtained from seeds show a wide phenotypic variability due to the heterozygous structure of the parental forms and the genetic recombination that occurs [19]. Seed propagation is also widely used to obtain the rootstocks on which the various varieties of *Rosa* are grafted [20]. At the Horticultural Research Station in Cluj-Napoca (HRS), affiliated to the University of

Agricultural Sciences and Veterinary Medicine of Cluj-Napoca, more than 40 new varieties have been obtained through artificial hybridization [21], and *Rosa canina* is usually used as a rootstock to produce grafted roses in the nursery. In Figure 4, different varieties of roses are grafted on *R. canina* at the HRS, where a collection of over 400 rose cultivars has been established in a rosarium. Both in rose breeding and in the production of roses grafted on *Rosa canina* seedlings obtained from seeds or from other species used as rootstocks, seed germination is of great interest, both for obtaining a higher percentage of seedlings and for their proper quality [19].

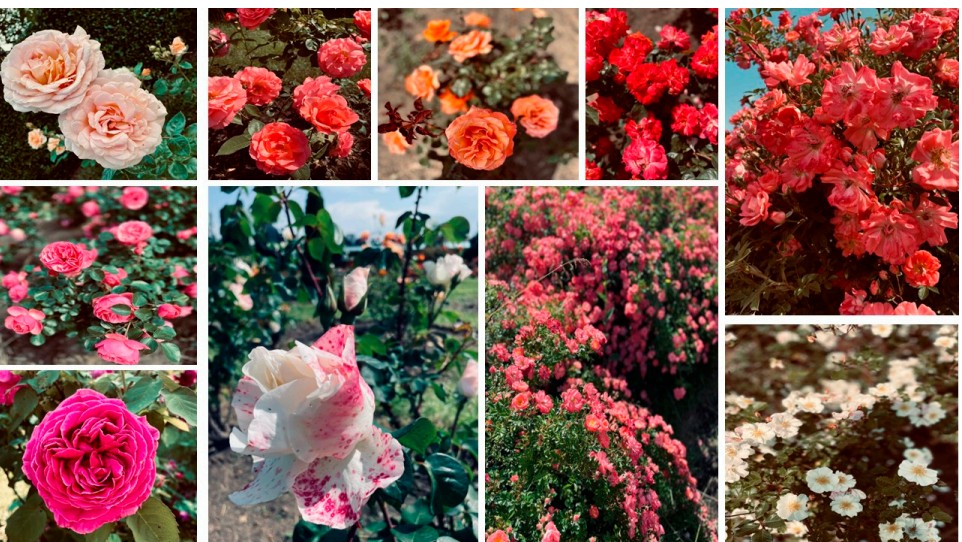

**Figure 4.** Different rose cultivars grafted on *Rosa canina* from the HRS rosarium Cluj-Napoca, Romania (photo by R.L.S.-D.).

In the practice of growing roses, there are many techniques of vegetative propagation. The most commonly used method is budding, carried out in August-September [18]. The use of biotechnological techniques allows obtaining uniformed clonal material from rootstocks for ornamental roses with the best parameters of inoculation zone compared with plants propagated by green cutting [22]. The grafted roses demonstrated winter hardiness, longevity, and high flower productivity [23].

## 2. Seed Characteristics and Germination within the *Rosa* Genus

Regarding higher plants' ability to survive as species, the seed stage is crucial. It is the plant's dispersal unit that may endure the time between the maturity of the seed and the establishment of the following generation as a seedling once it has germinated. The seed, which is primarily in a dry state for its survival, is well-equipped to withstand prolonged durations under unfavorable conditions. The seed goes into dormancy to maximize germination over time [24]. Additionally, dormancy stops pre-harvest germination. To comprehend how different environmental conditions and applied chemicals regulate germination, numerous investigations have been carried out. However, very little is still known about the process by which the rose embryo emerges from the seed to complete germination and how embryo emergence is blocked in dormant seeds [25]. Seed dormancy has been defined as the incapacity of a viable seed to germinate under favorable conditions [25,26]. In roses, hips typically contain between one and thirty seeds, and hip set and seed germination are frequently less than 50% [27]. In the breeding of cut and garden roses, mature hips are collected in late September and early October, or 3–4 months following pollination. Hips are used to collect and count seeds. After that, seeds are stored for roughly 6–12 weeks at 2–5 °C and a moderate moisture level. Seeds can be sown directly in the germination substrate for vernalization, depending on the environment; otherwise, preservation in a cold chamber is required. A modern cut rose breeding program is thought

to get off to a strong start with 120,000 seedlings after germination. Seeds are often sown on germination beds or benches at a density that can range from 150 to 400 seeds per square meter, depending on the available space and the number of seeds to start with. After two months, the vernalization period is deemed to be over [28]. Seeds of the *Rosa* genus are typically small, irregularly shaped (4.5–6 mm long), and vary in color depending on the species and variety (Figure 5). They are enclosed within a protective seed coat, which can range from smooth to slightly textured. *Rosa* seeds are generally characterized by their hard seed coat, which contributes to seed dormancy. The pericarp epi-, meso-, and endocarp layers make up a rose achene; the last structure is extremely impervious to water absorption (unpublished data). Therefore, the endocarpic layer may serve as a tegument-type physical barrier to achene germination [17].

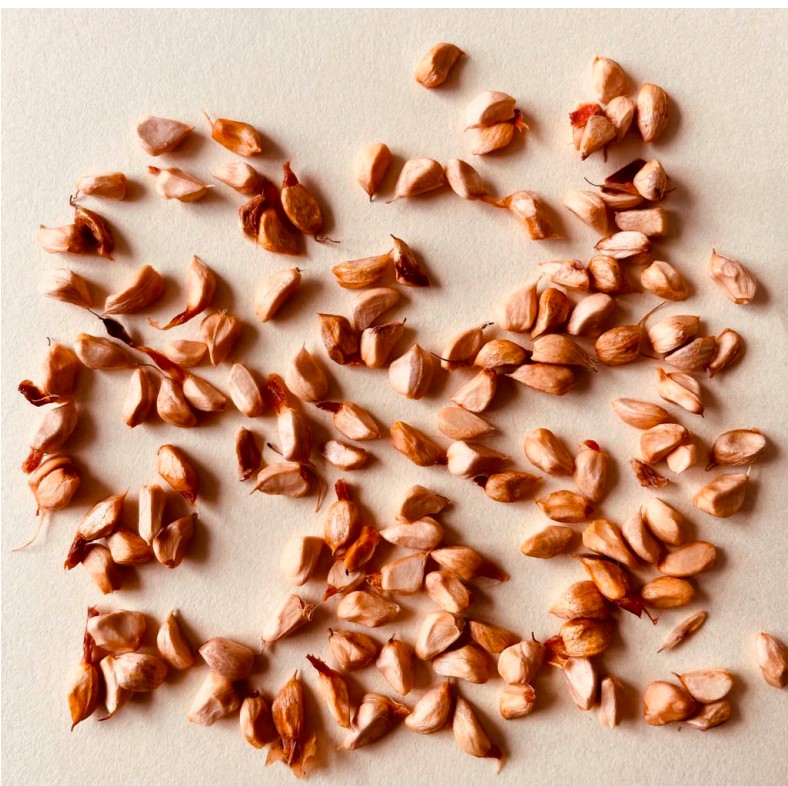

**Figure 5.** *Rosa canina* seeds. The seeds were obtained from fruits collected from the spontaneous flora of Transylvania, northwestern Romania (photo by R.L.S.-D.).

The seed coat protects the embryo but can present problematic challenges for germination. However, *Rosa* seeds also possess the potential resources for germination and subsequent growth into mature plants when provided with the appropriate environmental conditions and treatments to overcome dormancy [29]. Seed germination within the *Rosa* genus is a crucial process that determines the successful establishment and propagation of further obtained plants. The germination of *Rosa* seeds is often influenced by factors such as seed dormancy and is strongly influenced by environmental conditions and seed treatments. Many *Rosa* species exhibit various types of seed dormancy, including physical dormancy caused by hard seed coats, as well as physiological dormancy due to internal mechanisms [30].

An achene is where rose seeds are created. This dry fruit has a solitary seed that almost completely fills the pericarp. Achenes and seeds are frequently used interchangeably in literature and breeding. Gudin [27] analyzed rose seed propagation from the perspective of generative multiplication, as is the case for the creation of perfume (*R. gallica* and *R. × damascena*), for the propagation of rootstocks (Caninae rose), and for landscaping. According to the winter environment in the rose's native regions, a specific, adapted seed

dormancy was established. In species like *R. rugosa*, *R. gallica*, *R. canina*, and *R. soulieana*, it is known that dormancy can only be broken following intervals of chilling at 1–4 °C. Before germination, some roses require a second winter vernalization [31]. In contrast, the seeds of *R. persica*, which are adapted to the deserts of the Far East, do not respond to cold but can withstand dehydration. Depending on the rose species, there are differences in the period, kind, and cooling temperatures needed to break dormancy [27].

Overcoming seed dormancy often requires specific treatments, such as cold stratification or scarification, to break the inactivity process until favorable conditions are assured and promote germination. Nevertheless, abiotic factors like temperature, moisture, light, and substrate play important roles in the germination process. Understanding the germination requirements and dormancy mechanisms in the *Rosa* genus is essential for effective cultivation and conservation strategies for these valuable plants [17]. Roses are one of the most significant commercial crops in the world because of their significance as a decorative plant in landscape gardening, their high regard as a medicinal plant and for human nourishment, and the fact that the species is well adapted to a wide range of habitats [32,33]. Traditionally, stem cutting, layering, budding, grafting, and tissue culture are the primary vegetative methods used to propagate roses [34]. All of these techniques come with a number of issues, including a lack of rootstocks and a longer production period. The breeding of new cultivars, the restoration of native plants, the selection of rootstocks, and, in some varieties, the production of rose hips are all achieved through seed propagation; however, this process is challenging due to the low germination percentage that results from the prolonged seed dormancy [35–37].

According to Jackson and Blundell [38], Densmore and Zasada [39], and Bo et al. [37], inhibitors in the pericarp and testa, the hard pericarp, and physiological barriers in the embryo may be the main causes for the dormancy and delayed germination in rose achenes. Although it occasionally limits complete imbibition, the pericarp is permeable [36,37,40]. Some rose achenes experience dormancy as a result of a barrier in the shape of a hard pericarp [41], although this is not the only source of the condition [35,42].

The endocarp thickness in a rose achene pericarp can determine germination, as demonstrated by Gudin et al. [43]. Environmental factors, particularly temperature during achenes' maturation, and genetic factors, likely through their impact on the rate of embryo growth, regulate its thickness. The pericarpic tissues in their crosses, when only the male parent was different, were of the same maternal origin, whereas the embryos were of distinct hybrid origins; as a result, the endocarp (as the layer that is closest to the embryo), may be a crucial factor in determining an achene's ability to germinate [17].

High levels of abscisic acid (ABA) have also been found in the testa and pericarp of rose achenes, according to Bo et al. [37], which may prevent germination. The achenes' embryos have been shown to be fully formed and devoid of any morphological dormancy [37,38]. In addition, cold stratification has been used to get around physiological obstacles to embryo germination in a number of rose species [35,39,42].

Since only a few species of rose achenes have been studied, the process of dormancy in these plants is a complicated issue. Therefore, successful rose seed multiplication would benefit from better knowledge of dormancy in rose achenes. Only when the dormancy is broken can rose seeds have a better germination rate. Current efforts to break the dormancy have centered on two strategies: (a) removing the mechanical barrier known as the pericarp, which limits the embryo's growth and access to water and air; and (b) shortening the time the embryo must spend after ripening [44].

Achenes have not responded well to treatments that include soaking them in concentrated $H_2SO_4$ (sulphuric acid), exposing them to 100% oxygen-rich environments, dry storage, or cold stratification alone [44,45], or to various chemicals (such as $GA_3$). However, combining various therapies, such as $H_2SO_4$ scarification and cold stratification [39–41] or warm and cold stratification [39,42,45], can significantly enhance germination. To stimulate germination, several pretreatments have varying degrees of effectiveness depending on the species [41,45].

To successfully propagate horticultural plants, it is crucial to understand the specific kind of seed dormancy; however, at the moment, the majority of papers on seed dormancy do not specify the type of dormancy that was examined [46]. Researchers may not have looked at the many types of seed dormancy because there is no generally recognized system for defining dormancy. Morphological, physiological, morphophysiological, physical, and combinatorial dormancy are the five categories of dormancy that Baskin and Baskin [47] suggest as a new classification system for seed dormancy.

The definition of these various classes of dormancy is based on a number of characteristics, such as the embryo's morphology (underdeveloped or fully developed), the permeability of the seed coat to water (impermeable or permeable), and the physiological reactions of whole seeds to temperature or to a sequence of temperatures. With the use of such a classification system for seed dormancy, it is now possible to identify the type of dormancy by examining how different pretreatments affect germination [48].

## 3. Treatments Used to Stimulate Seed Germination in Species of the Genus *Rosa*

### 3.1. Inhibitory and Stimulatory Factors of Seed Germination in the Genus Rosa

In the generative propagation of *Rosa*, there are several factors that can impact germination negatively or positively. The low germination rate of *R. canina* seeds can be attributed to several factors, including:

- Seed dormancy: *R. canina* seeds often have seed dormancy mechanisms that prevent immediate germination. Seeds' dormancy can be caused by the hard seed coats or the presence of inhibitory substances that need to be broken down or leached out before germination can occur [21,23].
- Scarification requirements: Some *R. canina* seeds may require scarification, which is the mechanical treatment applied to the seed coat, to break dormancy. Scarification methods include nicking or sanding the seed coat or subjecting it to hot water or acid treatment within chemical treatments [21,49].
- Light requirements: Some *R. canina* seeds have light sensitivity and require exposure to light so that germination occurs. Seeds that require light will have low germination rates in darkness, so light can act as a germination stimulant for *Rosa* [49].
- Environmental factors: *R. canina* seeds may have specific environmental requirements for germination, such as moisture, temperature, and oxygen availability. If these conditions are not optimal, germination rates can be reduced [50].

Among the factors that can improve the germination of *R. canina* seeds, the most important are the following:

- Stratification: Many *R. canina* seeds require a period of cold stratification to break dormancy. Cold stratification involves subjecting the seeds to a period of cold, typically around 4 °C, for a certain duration. This mimics the natural conditions that the seeds would experience during the winter and helps overcome dormancy [51].
- Scarification: Some *R. canina* seeds have hard seed coats that can inhibit germination. Scarification techniques, such as mechanical scarification (e.g., nicking or scratching the seed coat) or chemical scarification (e.g., acid treatment), can help break the seed coat and enhance germination [52].
- Light exposure: While some *R. canina* seeds require light for germination, others may germinate better in darkness. Understanding the light requirements of specific seed lots can help optimize germination conditions accordingly [53].
- Moisture and temperature: Providing the seeds with adequate moisture and maintaining a suitable temperature range can promote germination. *R. canina* seeds generally require a moist environment, but excessive moisture can lead to fungal or bacterial issues. Optimal temperatures for germination typically range from 15 °C to 25 °C [54].
- Seed quality and age: Using high-quality, fresh seeds can improve germination rates. Older seeds may have reduced viability and lower germination rates [50].

Next, the main physical-mechanical, chemical, and biological treatments that can be applied to *Rosa* seeds to increase the germination rate will be presented, and a summary of the main methods, procedures, and techniques used to stimulate *Rosa* seed germination is shown in Table 2.

**Table 2.** Possibilities of increasing the germination percentage in the *Rosa* genus.

| Method, Procedures and Technics | Details | Germination | Reference |
|---|---|---|---|
| Scarification | Grinding (1, 5, 10 min) | 30% | [55] |
| Scarification | $H_2SO_4$ 97% (1, 5, 10 min) | 30% | [55] |
| Scarification | $H_2SO_4$ 50% (30 and 60 s) | >30% | [52] |
| Scarification | $H_2SO_4$ (2, 4, 6 h) | No germination | [48] |
| Scarification | NaOCl | 65.9% | [28] |
| Scarification | Fully removed testa | 39% | [48] |
| Microorganism | Inoculation in *Klebsiella oxytoca* C1036 | 50% | [55] |
| Microorganism | Inoculation natural microflora | 3% | [56] |
| Microorganism | Inoculation natural microflora + GarottaTM | 95% | [56] |
| Stratification | 8 weeks at 2.8 °C on moss | 37.1% | [54] |
| Stratification | Dry storage 68 w + cold stratification 16–24 w | 72–79% | [48] |
| Stratification | 11 w warm stratification + cold stratification | 13–18% | [51] |
| In vitro | $GA_3$ + manual scarification + agar medium + warm/chilling—cold/dark 21 days | 80–85% | [57] |
| $GA_3$ | 2000 ppm $GA_3$ for 12 h (greenhouse) | 74% | [30] |
| $GA_3$ | 300 ppm $GA_3$ for 24 h (Green house) | 24.7% | [30] |
| $GA_3$ | Chilled seeds + 200 ppm $GA_3$ for 6 h | 11.7% | [30] |
| $GA_3$ | Unchilled seeds + 400 $GA_3$ for 12 h | 52% | [30] |
| $GA_3$ | Stratification + pre-sowing $GA_3$ | 77.6 | [28] |

### 3.2. Physical/Mechanical Treatments

Various studies have been undertaken over the years on the species, many of them regarding the germination process. The scientific literature presents a low germination rate, often not more than 30%, which makes researchers all over the world have difficulties finding the proper methods to enhance seed germination within the *Rosa* genus [55].

- Harvesting time/period: For best seed quality, rose hips should be harvested when the fruits are mature, ripe, but firm [58]. Despite showing dormancy in the first year after fruition, the seeds typically germinate in the second year [59]. However, different harvesting periods have an impact on germination, with a previously documented 13–60% difference in germination frequency, with the best interval for harvesting rose hip being from late September until the beginning of October [60].

- Storage of seeds: In the study of Hoşafçı et al. [30], which aimed to investigate the effect of gibberellic acid ($GA_3$) treatments on the germination of dog rose seeds, it was discovered that even the seed storage type can influence the germination rate. The three pre-treatments used—(1) seeds kept at 4 °C; (2) seeds in fruit kept at 4 °C; and (3) seeds in fruit kept at room temperature (25 °C)—had considerable effects on the germination of dog rose seeds in the field experiment. The lowest (all-over mean) rate of germination (22.5%) was obtained for seeds kept in the refrigerator. The highest (all-over mean) rate of germination (39.1%) was obtained for seeds kept inside fruits at room temperature [30].

- Scarification: While most vegetable seeds germinate readily upon exposure to normally favorable environmental conditions, many seed plants that are vegetatively (asexually) propagated fail to germinate readily because of physical or physiologically imposed dormancy. Physical dormancy is due to structural limitations for germination, such as hard, impervious seed coats. Under natural conditions, weathering

for a number of years weakens the seed coat. Certain seeds have a tough husk that can be artificially worn or weakened to render the seed coat permeable to gases and water by a process known as scarification [61]. Scarification is a process that includes modifying, weakening, or opening a seed's covering in order to promote germination. Scarification frequently involves mechanical, thermal, and chemical processes. Many plant species have seeds that are frequently resistant to water and gases, which delay or prevent germination [48].

Scarification techniques, such as mechanical scarification (e.g., nicking or scratching the seed coat), as well as subjecting the seeds to hot water, can help break the seed coat and enhance germination [61]. According to Zhou et al. [48], scarified achenes have somewhat higher imbibition rates than intact achenes. However, the germination rate was only about 30% overall, which is a moderate improvement. The damage to the seeds during the grinding process may result in a slight reduction in germination. In a study by Lee et al. [55], several treatments were used in order to enhance germination in *Rosa* seeds. They used physical scarification. Grinding the seeds for 1, 5, and 10 min (+control) and stratifying them at 4 °C/12 weeks resulted in a germination percentage of not more than 30%; overall, there was no significant improvement. Like in the study of Zhou et al. [48], the fact that some of the seeds were damaged during the grinding contributed to the low germination rate. Pippino et al. [28] reported experiencing both chemical and microbiological seed scarification. Prior to stratification, a light scarification with sodium hypochlorite (NaOCl) encouraged more uniform germination and raised the percentage of germination from 49.2% to 65.9%.

UV-Irradiation: UV-B irradiation has some activity on the surface of the seed cells, stimulating rapid germination [62]. In a study by Lee et al. [63], UV-B radiation was applied to *R. rugosa* seedlings for 5, 10, and 30 min. Before 10 min of irradiation, there was no noticeable difference between the treated and untreated seedlings. Instead, the germination drastically decreased after 30 min of exposure. The inner cells of the seed appeared to be harmed by excessive UV-B exposure. In the germination study [62], such an effect was more or less expected. Finding the homeosis stage will be crucial to enhancing germination, but it will not be realistic due to the varied pericarp thickness of each seed caused by different cross-combinations. Living cells are susceptible to being harmed by UV-B. Future research should examine additional UV 185 sources [64]. When grown with the embryos facing the sun, half-cut achenes could germinate at a maximum rate of 100% in less than a week. Regardless of cold storage, half-cut achenes at 90 DAP germinated 100% of the time. To mimic the appearance of different light colors, various LED lights (red, blue, yellow, green, and white) were lit over the half-cut achenes. Ninety percent of *R. rugosa* seeds grew under blue LED lighting in a week of culture, and those seedlings afterwards had the best growth and development. Half-cut achenes were discovered to be a useful technique to enhance seed germination in *R. rugosa* without stratification or scarification. Using this method, rose cultivar breeding might be examined [63].

- Gamma ray irradiation: To remain competitive in the flower industry, breeders are constantly striving for fresh and innovative varieties. Low seed germination rates continue to be a significant issue in conventional rose breeding projects. To improve seed vigor and productivity and to improve the sprouting and emergence of buds that are carried out by seed coating, mutagenic agents, such as ionizing radiation, may be utilized. Ionizing radiation treatments must be tailored for each unique cultivar since the effects of radiation on seed vigor are typically genotype-dependent [64]. In a study by Giovannini et al. [64], hybrid tea rose seeds were exposed to gamma rays (0, 50, 100, and 200 Gy) in order to establish a radiation regimen for boosting seed germination. *Rosa* hybrid commercial cultivars' six different crossings' seeds were used to collect data on germination capacity and radio-tolerance. The final germination percentage and germination energy of the seeds, regardless of the cross, were not significantly affected by the range of gamma ray doses evaluated. These findings contrast with research done on seeds of various species, including *Vigna unguiculata* L. [65], *Citrus*

*jambhiri* Lush. [66], and *Withania somnifera* L. [67], in which a gradual decline in seed germination and seedling vigor from lower doses to higher doses in given treatments of gamma rays was found. Since there are other findings that show that ionizing radiation has positive impacts on seed vigor, this is a contentious topic.

- Medium: Anderson and Byrne [68] tested the influence of stratification media and genotype over the germination process. They stratified fresh rose seeds from different hybrids for 10–12 weeks at 2.8 °C in milled sphagnum moss, sand, perlite, vermiculite, and moist filter paper. The achenes germinated the best on moss. When placed on moist milled sphagnum moss or agar and stratified for 8 weeks at 2.8 °C, once again, the best germination was on moss. The germination of the genotypes varied greatly, ranging from 0.7 to 37.1% [64]. According to Carter [69], rose seed germinated more effectively after stratification in damp peat moss or sand as opposed to moist vermiculite. The effectiveness of sphagnum moss as a stratification medium was demonstrated in their experiment. The outcomes provided more proof that the medium is essential for seed germination. The smallest rate of germination was encouraged by stratification on filter paper, while the best germination was obtained with sphagnum moss. In terms of stimulating germination, perlite fell somewhere between sphagnum moss and the other two media (sand and vermiculite). Sphagnum moss performed better or on par with the other stratification media across all genotypes examined. In contrast to Yambe et al.'s [70] findings for *R. multiflora*, leaching of WOB-28 rose seed for three or more days significantly reduced germination. Such a divergence may have been caused by changes in leaching technique, genotypic effects, or leaching water temperature.
- Stratification: Aches treated with warm and cold stratification were used to examine the effects of temperature and water stress. Freshly harvested achenes are devoid of any physical, morphological, or morphophysiological dormancy since the pericarps are permeable and the embryo is completely formed. Despite softening or even completely removing the pericarp, the germination percentage remained low (5%), while completely removing the testa greatly increased germination (39%), indicating the potential existence of germination inhibitors in the testa [52]. According to Zhou et al. [52], dry storage for 68 weeks followed by cold stratification for 16 or 24 weeks resulted in maximum germination (72–79%) in *R. multibracteata*. The most popular method for removing rose seed dormancy is chilling, as most species' achenes will eventually germinate if cooled for an extended length of time. For certain species, cold stratification periods equivalent to one field winter are sufficient. Only if the temperature of warm incubation was too high did the interruption of chilling with warm incubation result in subsequent dormancy induction. The dormancy of the seeds did not change when they were kept below this "compensating" temperature, and they may accrue the effects of chilling despite warm breaks [71].

The results of a study by Alp et al. [51] evaluated the effects of various warm stratification durations on the seed germination of some *Rosa* species, including *R. heckellana* ssp. *vanheurckiana*, *R. pulverelanta*, *R. dumalis*, and *R. canina* naturally grown. The seeds were stored for 10, 11, and 12 weeks at 25 °C before cold storage at 5 °C until the start of germination testing. In terms of germination, *R. heckellana* ssp. *vanheurckiana* seeds reacted to treatments differently than seeds of other species. When the seeds were stored in warm stratification followed by cold stratification, germination occurred promptly (between 1 and 3 weeks). The other species' seeds need 5 months of cold stratification after warm stratification in order to emerge from hibernation. The overall germination percentages were 13.80% in *R. pulverelanta*, 18.80% in *R. canina*, and 13.53% in *R. dumalis* at 25 °C of warm stratification and 5 °C of cold stratification. For these three taxa, an 11-week warm stratification period followed by a cold stratification period was determined to be the most successful stratification method [50].

- Temperature: Temperature is an important environmental component that inhibits seed germination. A hallmark of rose (*R. canina* L.) seedlings is the physical and

physiological dormancy, which is often disrupted during warm weather, followed by cold stratification. When prepared seeds were subjected to a temperature of 20 °C, secondary dormancy was produced. A study by Pawowski [72] sought to discover and functionally describe the proteins linked to rose seed dormancy management. Using 2-D electrophoresis, proteins from primary dormant, following warm and cold stratification (nondormant), and secondary dormant seeds were examined. Mass spectrometry was used to determine which proteins were abundant in different ways.

The findings indicated that secondary dormancy was associated with more common spots than warm stratification, and that cold stratifications had the greatest impact on spot variability. The rise in actin and mitochondrial proteins at the end of dormancy suggests modifications to cellular processes and seed germination preparation. Low quantities of legumin, metabolic enzymes, and actin were found in secondary dormant seeds, indicating the use of storage resources, a decline in metabolic activity, and cell elongation [72]. When rose seeds were brought out of dormancy, more cellular and metabolic proteins that encourage germination were present in greater quantities. These proteins were reduced, and germination was arrested as a result of the induction of secondary dormancy. Both high temperature and water stress lowered germination in achenes treated with warm plus cold stratification [52].

The physical and physiological dormancy of rose (*R. canina* L.) seed is broken by warm, then cold stratification. Secondary dormancy is induced in pretreated seeds when they are exposed to a temperature of 20 °C. When rose seeds were brought out of dormancy, more cellular and metabolic proteins that encourage germination were present in greater quantities. These proteins are reduced, and germination is arrested as a result of the induction of secondary dormancy [72].

- Tetrazolium staining is typically used to assess the quality of rose seeds due to the vast variety of dormancy-breaking needs within each species [73]. The first step is to soak the achenes in water for 24 h. The pericarp is broken open by applying firm pressure with a knife. After the testa has been scratched or snipped at the cotyledon end, the seed is submerged for 6 h at room temperature in 1% tetrazolium chloride. The testa is slit along the side, and the embryo, which fills the seed cavity, is squeezed or teased out for evaluation [74]. The excised embryo method may also be used, although it has little advantage over tetrazolium staining [73]. For purposes of determining fill and chalcid infestation levels, x-radiography is suitable [74]. X-radiography is helpful for assessing the amount of fill and chalcid infestation [74].

### 3.3. Chemical Treatments

- Sulphuric acid: To boost ingestions and break physical dormancy, sulphuric treatment of the rose pericarp is advised rather than scarification. However, due to the uneven thickness of the rose pericarp, the application of sulphuric acid necessitates substantially more stabilization. This kind of scarification technique has been employed in certain research on roses, although it often only applies to a small number of cross-combinations or particular species [52,63,74]. Zhou et al. [48] treated seeds with sulphuric acid for 2, 4, and 6 h and observed no germination. The overall rate of germination was 30%. The treated and untreated samples did not differ significantly from one another. The longer the sulphuric acid treatment, the smaller the rate obtained. It appears that even during this brief course of treatment, sulfuric acid could harm the embryo [63].

Soaking the seeds in sulphuric acid (97%) for 1, 5, and 10 min (+stratification 12 weeks/4 °C) resulted in no more than a 30% germination rate [55]. Apparently, sulphuric acid can damage the embryo in a very short period of time. According to Younis et al. [75], applying $H_2SO_4$ 50% for 30 and 60 s gives better germination. In a study by Zhou et al. [52] no germination was observed after the seeds were treated with $H_2SO_4$ for 2, 4, and 6 h. According to Roberts and Shardlow [76], a sulphuric acid treatment before warm + cold

stratification can enhance germination. Cullum et al. [77] suggest that acid scarification can be eliminated all along, and the warm stratification time should shorten if the achenes warm stratification is made with compost activator. When paired with cold stratification, dry storage, scarification with $H_2SO_4$, and warm stratification increased germination and reduced the time needed for cold stratification to break dormancy [52]. Sulfuric acid treatment prior to warm and cold stratification promoted germination in nursery propagation of the rootstock rose *R. dumetorum* (*R. corymbifera*) 'Laxa' [76]. If the achenes are heated-stratified with compost activator, the acid scarification can be avoided, and the warm stratification duration decreased [77]. It appears that the purpose of these treatments, whether through acid or microbial digestion, is to weaken the pericarp at the sutures. Growth hormones like gibberellic acid or benzyladenine can be vacuum-injected into the achenes, as for *R. dumetorum* 'Laxa', to increase their response to warm and cold stratification, which suggests that something other than simple mechanical restriction may be at play [77]. Similar to this, it is possible to encourage the achenes of the relatively non-dormant multiflora rose to germinate without chilling by treating them with enzymes that weaken the pericarp sutures or by leaching the incubation solution with activated charcoal to remove any inhibitors [71].

Gibberelic acid: Neither gibberellic acid ($GA_3$) nor 'smoke water' (water through which smoke had been bubbled for 2 h) had any positive effect on germination, even on seeds that had been mechanically stratified or scarified [52]. According to Hoşafçı [30], there is a positive correlation between the concentration of $GA_3$ applied and the germination rate. He obtained 74% germination in greenhouse from seeds treated with 2000 ppm $GA_3$ for 12 h. A lower germination rate (24.7%) was observed when seeds were treated with 300 ppm $GA_3$ for 24 h. The same authors' field study found that germination varied from 11.7% for chilled seeds treated with 200 ppm $GA_3$ for 6 h to 52.0% for unchilled seeds treated with 400 ppm $GA_3$ for 12 h and stored inside fruits at room temperature [30]. Pre-sowing treatments with $GA_3$ were carried out following stratification. When compared to immersion in water (respectively 64.8%, 32.8 days), stratified seeds immersed in 1 g $L^{-1}$ $GA_3$ had a significantly higher percentage of seed germination (77.6%), mean germination time (26.0 days), and uniformity of germination (2.5%) [28]. According to Meyer [71], in a non-dormant rose, the achenes might be stimulated into germination without chilling, either with macerating enzymes or by leaching with activated charcoal. Using macerating enzymes to break dormancy, Yambe et al. [70] demonstrated phytochrome-mediated light requirements. Acid scarification is reported to substitute for warm pretreatment in the cultivated rose, *R. gallica* L. [78]. Other closely related species, including *R. arvensis*, *R. floribunda*, *R. foetida*, *R. jundzillii*, *R. micrantha*, *R. obtusifolia*, and *R. rubiginosa*, have also been noted to enter a dormant state [60]. For *R. canina* seeds maintained in the refrigerator for 180 days, various treatments, including stratification and scarification with $GA_3$ and $H_2SO_4$, have been reported to remove dormancy and resulted in a 6.2% germination rate within the first year [60]. By employing solely $GA_3$ treatment, no germination was obtained in any of the examined species [60].

- Nitric acid: Younis et al. [75] tried to break achenes dormancy by using nitric acid. Nitric acid is a nitrogen oxoacid that has a role as a protic solvent and reagent. As is the case with sulphuric acid, it was used in order to break the seed's pericarp. Nitric acid (65%) with different time exposures (30, 60, and 90 s) was tested. Although it was expected, no significant effect of the acid was observed [78].

### 3.4. Biological Treatments

- Microorganism Treatment: Effective microorganisms (EM) are used to facilitate seed germination [79]. Microorganism: *Klebsiella oxytoca* C1036 is a strain that has been identified to act against the soft-rot pathogen in Tabacco [80]. Lee et al. [55] used this strain to enhance germination. The seeds were immersed in a suspension of the strain for 1 and 48 h. The germination rate rose to 50% with longer immersion. Treated seeds

germinated twice as well as those of non-treated seeds. Other useful microorganisms should be investigated in further studies [55].

*Rosa corymbifera* 'Laxa' germination under typical conditions was no more than 2%. This was accomplished when the natural microflora on the seeds was present. The hips are where the microflora first appeared, and when the seeds are extracted, they are vaccinated. No germination occurred when microorganisms were not included in these pretreatments. Three percent of surface-sterilized seedlings that were then inoculated with natural microflora grew. GarottaTM, a commercial compost activator, was added to the industrial pretreatment to boost germination to 95% [56].

Remedier®, a combination of *Trichoderma harzianum* and *Trichoderma viridae*, and EmercalTM, bacteria and co-metabolites produced by bacterial fermentation, when added to the stratification sand, increased the proportion of germinated seeds but had no role in germination uniformity [28].

In a study by Kazaz [80], an investigation was undertaken to determine how microbial inoculation affected *Rosa damascena* Mill germination and the breaking of seed dormancy. The seeds underwent a 150-day cold stratification period at $4 \pm 1$ °C after 4 weeks of warm stratification at 25 °C. The seeds were injected with four different microbial fertilizers, including EM•1®, B:speelTM, BioplinTM, and PhosfertTM, prior to stratification. In the study, the treatments for microbial inoculation considerably ($p < 0.01$) increased the percentage of early germination during cold stratification. The EM1® yielded the highest rate of premature germination during stratification (69.3%). In terms of cumulative germination percentage, EM•1® (100.0%) had the highest germination rate, followed by PhosfertTM (84.0%) and B: seepelTM (84.0%), while the control treatment (69.3%) had the lowest germination rate. In comparison to the control, the EM•1® decreased the mean germination time by 1.7 days. In conclusion, it was found that dormancy was broken and germination was greatly enhanced with microbial inoculation (especially EM•1®) of oil rose seeds and a stratification time of 150 days [81].

- Macerating enzymes: *R. multiflora* Thunb. achenes were treated for 36 h with 1% Driselase, a macerating enzyme, which significantly enhanced the germination percentage. When the achenes were exposed to the enzyme for a longer time, the seeds germinated more quickly. In comparison to Driselase, treatment with Cellulase Onozuka increased seed germination at a lower dosage. Pure pectinase and cellulase preparations had outcomes that were comparable to those of the mentioned enzyme treatments. Pectinase treatment was more effective than cellulase treatment. These enzymes probably made the pericarp's suture less rigid, which caused the pericarp to split [82].
- Genetic make-up of the seed: One important factor contributing to poor germination is the genetic make-up of the seed. This is the reason many breeders maintain thorough records of seed set and germination in order to select the genitors that produce the greatest number of offspring. Techniques for treating seeds, such as harvesting, stratification, scarification, and leaching, are essential to maximizing germination [57].
- Break seed dormancy under in vitro conditions: In the study by Hajyzadeh [57], it was aimed to break the seed dormancy of rose hip under in vitro conditions by applying multiple strategies in an efficient manner. The seeds were given various doses of GA$_3$, mechanically scarified, stratified on agar-solidified MS medium containing GA$_3$ alone or in combinations with the suggested treatments, and then given a controlled physiological treatment that involved alternately giving the seeds warm/chilling and cold/dark treatments for 21 days, followed by 18 days of warm/light treatments.

It was found that the rosehip seeds might germinate in various ways depending on whether the scarified seeds were spread over an agar-solidified MS medium with or without GA$_3$. The best seed germination (80.00–85.00%) was observed when the three treatments were combined and the seeds were given controlled and alternately warm and cold treatments for 21 days, leaving them for 18 days in warm/light conditions. These



crucial insights could be used in breeding and propagation operations to develop novel rosehip rootstock and fruit varieties [57].

**4. Conclusions**

Seed germination in the *Rosa* genus is a complex process influenced by various factors. Our understanding of these factors is crucial for successful cultivation, conservation, and propagation of *Rosa* species. Optimal conditions for germination vary among species, highlighting the importance of understanding the specific requirements for each *Rosa* species of interest. Advances in research techniques, such as in vitro germination and molecular approaches, have contributed to a deeper understanding of *Rosa* seed germination biology. Further research is needed to explore the germination ecology of specific *Rosa* species, uncover additional dormancy mechanisms, and develop efficient germination protocols for propagation purposes. The knowledge gained from studying *Rosa* seed germination can aid in the conservation of endangered species, the selection of suitable cultivars for horticulture, and the development of sustainable rose production systems.

**Author Contributions:** Conceptualization, R.E.S. and R.L.S.-D.; methodology, R.L.S.-D.; software, R.L.S.-D.; validation, R.E.S., A.F.S. and C.D.; formal analysis, A.M.T.; investigation, I.M.M.; resources, G.R.; data curation, R.L.S.-D.; writing—original draft preparation, R.L.S.-D.; writing—review and editing, R.E.S. and C.D.; visualization, R.E.S. and A.F.S.; supervision, R.E.S.; funding acquisition, R.L.S.-D. All authors have read and agreed to the published version of the manuscript.

**Funding:** This research was partially funded by the University of Agricultural Sciences and Veterinary Medicine of Cluj-Napoca (USAMVCN).

**Data Availability Statement:** Not applicable.

**Acknowledgments:** The research was carried out with the partial support of the Doctoral School of the University of Agricultural Sciences and Veterinary Medicine of Cluj-Napoca (USAMVCN) for R.L.S.-D.—student.

**Conflicts of Interest:** The authors declare no conflict of interest.

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
