# Peer review of "Seed Germination within Genus Rosa: The Complexity of the Process and Influencing Factors"

_horticulturae, doi:10.3390/horticulturae9080914_

Round 1
Reviewer 1 Report
The botanical properties and importance of Genus Rosa were introduced in this review firstly, then the seed germination and treatments were reviewed. It provides a better understanding for readers about this plant, so it contains some valuable informations. However, some questions should be clarified, and the details of comments as below:
1. The title, it is not generalized. From 1-3 section, the authors just introduce the characteristics and botanical aspects, they didn't show from the current title. So please consider it.
2. About the writing, some places should be corrected. For example, line 43, the details of vitamin don't need to list, what else vitamin except for ABDEK? The contribution of authors don't need to note in the Figure.
3. Line 558, it should be 5.2, and the conclusion section is for 6.
4. Reference list, the format of reference is not uniform, just for example, the place of Year, and the difference totally in No. 54, so please double-check the format one by one. It must be followed the guideline for authors of Journal.
5. What is the function for figure 2 and 3? The organization of this article should be re-consider. What is the consideration for too much introduction of section 1-3?
Author Response
Dear reviewer,
Thank you very much for all the helpful insights that you gave regarding the manuscript. It really helped improve the paper a great deal. I really hope that I was able to accomplish all the requests suggested. Overall, the paper is very much improved and I really hope it will be accepted for publication.
Once again, many thanks!
Comments and Suggestions for Authors
The botanical properties and importance of Genus Rosa were introduced in this review firstly, then the seed germination and treatments were reviewed. It provides a better understanding for readers about this plant, so it contains some valuable informations. However, some questions should be clarified, and the details of comments as below:
- The title, it is not generalized. From 1-3 section, the authors just introduce the characteristics and botanical aspects, they didn't show from the current title. So please consider it. Response: first of all, many thanks for all the suggestions, they were well received. We did not change the title but we reorganized the entire manuscript.
- About the writing, some places should be corrected. For example, line 43, the details of vitamin don't need to list, what else vitamin except for ABDEK? The contribution of authors don't need to note in the Figure. Response: The author’s contribution from Figure 1 was deleted, as requested.
- Line 558, it should be 5.2, and the conclusion section is for 6. Response: we reorganized the manuscript and corrected the sections number.
- Reference list, the format of reference is not uniform, just for example, the place of Year, and the difference totally in No. 54, so please double-check the format one by one. It must be followed the guideline for authors of Journal. Response: the reference section was rechecked and re-done.
- What is the function for figure 2 and 3? The organization of this article should be re-consider. What is the consideration for too much introduction of section 1-3? Response: the organization of the paper was reconsidered and the section 1-3 shortened.
Reviewer 2 Report
Dear Authors:
Seed germination is a key stage in the plant life cycle, and in this study, the effects of different factors on the germination of Rosa canina as well as technical means to overcome seed dormancy are reviewed. The review shows that dormancy usually consists of physical dormancy induced by hard seed coat and physiological dormancy caused by internal mechanisms; special treatments such as low-temperature stratification, de-scarification, or chemical treatments promote germination, and environmental factors play a crucial role in rose germination.
Several suggestions were as follows:
(1) The paper uses a large number of pictures of plants of the genus Rosa, and each picture shown in the paper has its own expression, but I can't understand why it is necessary to show these plant pictures here; moreover, when there are more than one picture in a figure, it should be labeled with different "a", "b", etc. to differentiate them;
(2) The article provides an extended discussion of the complexity of the seed germination process and the factors that influence it in the genus Rosa, but on pages 1-9 the description, botanical characteristics, and chemical composition of Rosa are examined, which in my opinion is irrelevant to the topic of the article's research and does not need to be examined in detail;
(3) On pages 9-12, the paper presents an overview of the factors influencing seed germination in Rosa canina, but only lists the different influencing factors, but does not discuss in detail the mechanisms by which the factors influence seed germination;
(4) On pages 12-17, the paper presents an overview of treatments to promote seed germination of Rosa canina. Firstly, the discussion of factors is too confusing, and it is recommended to develop the discussion from three perspectives: physical, chemical and biological, instead of stacking them together in a messy manner; secondly, the discussion of each influencing factor should be centered on its importance and mechanism of action, instead of simply presenting the results of 1-2 pieces of literature of other people; and the influencing factor should be a common indicator of the Rosa spp. and should not just describe some specific treatments of specific roses spp;
(5) On pages 12-21, the article discusses the research on germination results in the genus Rosa, but I very much don't understand why this section, which is very similar to the factors affecting seed germination in Rosa canina, is being reorganized again by singling out the results of other people's research on factors of germination treatments, which should have been integrated and reorganized with the previous section;
(6) There are errors in the numbering of the title of the article, e.g., the title "2.1 Morphology of the Species" should not be needed; "6.2 Chemical Treatments", etc., are definitely not "5.2 Chemical Treatments"ï¼›
(7) There are some errors in the formatting of the text, such as the correct spacing of the content on page 21, lines 627-637, and it is recommended that this be double-checked;
(8) There are some problems in the references, and some of them are missing page numbers, which are suggested to be verified carefully.
The English language meets the requirements for journal publication, with no obvious grammatical errors.
Author Response
Dear reviewer,
Thank you very much for all the helpful insights that you gave regarding the manuscript. It really helped improve the paper a great deal. I really hope that I was able to accomplish all the requests suggested. Overall, the paper is very much improved and I really hope it will be accepted for publication.
Once again, many thanks!
Comments and Suggestions for Authors
Dear Authors:
Seed germination is a key stage in the plant life cycle, and in this study, the effects of different factors on the germination of Rosa canina as well as technical means to overcome seed dormancy are reviewed. The review shows that dormancy usually consists of physical dormancy induced by hard seed coat and physiological dormancy caused by internal mechanisms; special treatments such as low-temperature stratification, de-scarification, or chemical treatments promote germination, and environmental factors play a crucial role in rose germination.
Several suggestions were as follows:
(1) The paper uses a large number of pictures of plants of the genus Rosa, and each picture shown in the paper has its own expression, but I can't understand why it is necessary to show these plant pictures here; moreover, when there are more than one picture in a figure, it should be labeled with different "a", "b", etc. to differentiate them; Response: first of all, I would like to thank you for checking our paper and suggesting the improvements; they were well received. We consider that the pictures contribute to a better understanding of the presented information and gives a note of originality. The figures were noted with a, b, etc.
(2) The article provides an extended discussion of the complexity of the seed germination process and the factors that influence it in the genus Rosa, but on pages 1-9 the description, botanical characteristics, and chemical composition of Rosa are examined, which in my opinion is irrelevant to the topic of the article's research and does not need to be examined in detail; Response: thank you for your observation; we restructured and synthetized this part. We consider that is useful for the readers in order to have a better understanding of the subject.
(3) On pages 9-12, the paper presents an overview of the factors influencing seed germination in Rosa canina, but only lists the different influencing factors, but does not discuss in detail the mechanisms by which the factors influence seed germination; Response: we re-organized the entire manuscript and made a section dedicated to the mechanism by which the germination is influenced.
(4) On pages 12-17, the paper presents an overview of treatments to promote seed germination of Rosa canina. Firstly, the discussion of factors is too confusing, and it is recommended to develop the discussion from three perspectives: physical, chemical and biological, instead of stacking them together in a messy manner; secondly, the discussion of each influencing factor should be centered on its importance and mechanism of action, instead of simply presenting the results of 1-2 pieces of literature of other people; and the influencing factor should be a common indicator of the Rosa spp. and should not just describe some specific treatments of specific roses spp; Response: thank you for the recommendation, we re-structure the section as suggested: physical, chemical and biological treatments.
(5) On pages 12-21, the article discusses the research on germination results in the genus Rosa, but I very much don't understand why this section, which is very similar to the factors affecting seed germination in Rosa canina, is being reorganized again by singling out the results of other people's research on factors of germination treatments, which should have been integrated and reorganized with the previous section; Response: we re-organized accordingly, and integrated this part to the previous one.
(6) There are errors in the numbering of the title of the article, e.g., the title "2.1 Morphology of the Species" should not be needed; "6.2 Chemical Treatments", etc., are definitely not "5.2 Chemical Treatments". Response: We made a different structure for the paper, and corrected accordingly.
(7) There are some errors in the formatting of the text, such as the correct spacing of the content on page 21, lines 627-637, and it is recommended that this be double-checked; Response: the correction have been done. Thank you!
(8) There are some problems in the references, and some of them are missing page numbers, which are suggested to be verified carefully. Response: the reference section has been done from scratch.
Comments on the Quality of English Language
The English language meets the requirements for journal publication, with no obvious grammatical errors.
Reviewer 3 Report
Abstract section: please mention seed germination as a crucial step in Rose breeding by hybridization. It is inconceivable when the seed germination rate is low.
Please correct root stocks to 'rootstocks'. Line 69
Here also it would be beneficial to mention that seed germination is extremely important due to mass seedling utilization as rootstocks. Line 109: Please elaborate further, it is very important
The plant can grow new shoots from its long-lasting (perennial) rootstock, but it primarily reproduces through seeds.
These parts are relatively long but not relevant to the title and scope of the paper: 3.1. Chemical Composition of the Hips; 3.2. Health Benefits of Rosa
Line 177 'to graft into the eyes while sleeping in August', please correct to 'by budding'
Figure 4, please provide the cultivars' names.
Correct Like said before to 'as stated above' or something similar, use more academic expression.
Bullets under the lines 226 and 242 can be combined into 'problem' and 'solution', to avoid repetitions.
Line 320, fruition, you mean ripening?
Line 323: September until the beginning of October in the northern hemisphere.
Lines 334 - 338: this was already stated many times, please check the whole manuscript and delete repetitions.
Line 353: any progress is progress when it comes to the dormant seeds, so correct 'not a considerable improvement' to moderate improvement.
Half-cut achenes are very difficult to achieve, is this method justified?
Microorganism Treatment, this part deserves more attention.
You mainly refer to the Rosa rugosa, the title should be corrected accordingly.
Line 421, capital letter for 'this'.
The structure of the manuscript is not uniform, please correct it. Keep one style.
Table 2 is very beneficial.
Chapter 5.1 seem like mere repetition.
Line 578 is already stated in the line 492 for instance.
No comment.
Author Response
Dear reviewer,
Thank you very much for all the helpful insights that you gave regarding the manuscript. It really helped improve the paper a great deal. I really hope that I was able to accomplish all the requests suggested. Overall, the paper is very much improved and I really hope it will be accepted for publication.
Once again, many thanks!
Open Review
Comments and Suggestions for Authors
Abstract section: please mention seed germination as a crucial step in Rose breeding by hybridization. It is inconceivable when the seed germination rate is low.
Please correct root stocks to 'rootstocks'. Line 69. Response: thank you very much for all the suggestions, they were well received and helped improve the manuscript. The correction has been done.
Here also it would be beneficial to mention that seed germination is extremely important due to mass seedling utilization as rootstocks. Line 109: Please elaborate further, it is very important. Response: the suggestion has been takes into consideration.
The plant can grow new shoots from its long-lasting (perennial) rootstock, but it primarily reproduces through seeds. Response: this aspect has been taken care of, thank you!
These parts are relatively long but not relevant to the title and scope of the paper: 3.1. Chemical Composition of the Hips; 3.2. Health Benefits of Rosa. Response: the entire paper has been restructured. These chapters have been synthetized.
Line 177 'to graft into the eyes while sleeping in August', please correct to 'by budding' Response: Thank you very much for this observation, we corrected accordingly.
Figure 4, please provide the cultivars' names.
Correct Like said before to 'as stated above' or something similar, use more academic expression. Response: taken into consideration, thank you!
Bullets under the lines 226 and 242 can be combined into 'problem' and 'solution', to avoid repetitions. Response: this section has been modified.
Line 320, fruition, you mean ripening? Response: yes, thank you!
Line 323: September until the beginning of October in the northern hemisphere. Response: corrected accordingly.
Lines 334 - 338: this was already stated many times, please check the whole manuscript and delete repetitions. Response: the whole paper has been re-organized and the repetitions have been deleted.
Line 353: any progress is progress when it comes to the dormant seeds, so correct 'not a considerable improvement' to moderate improvement. Response: this aspect has been taken into consideration, thank you!
Half-cut achenes are very difficult to achieve, is this method justified? Response: the method might be hard to achieve, but the germination percentage was good.
Microorganism Treatment, this part deserves more attention. Response: the section has been improved, thank you.
You mainly refer to the Rosa rugosa, the title should be corrected accordingly. Response: with the new structure of the paper, there is no need. Thank you!
Line 421, capital letter for 'this'. Response: Done! Thank you!
The structure of the manuscript is not uniform, please correct it. Keep one style. Response: the paper has been reorganized and edited according to the journal’s template.
Table 2 is very beneficial.
Chapter 5.1 seem like mere repetition. Line 578 is already stated in the line 492 for instance. Response: Thank you for your recommendation! A very large part of the manuscript was restructured and re-written.
Comments on the Quality of English Language
No comment.
Reviewer 4 Report
According to the title of the review, the authors summarize the scientific knowledge on seed germination in the genus Rosa.
Generally, although the manuscript gives a relatively good overview of the subject of the title, it contains a long "introduction" of less relevant knowledge (distribution, uses, botanical description of Rosa canina, etc.). This section, which can be called an introduction, goes all the way to line 183 on page 9, and since it contains few information related to germination, it should be shortened to a maximum of 20-30 lines. In this part, by the way, it is not quite clear (and sometimes highlighted in the later text) whether the aim was the examination really of /Rosa canina/ or the whole /Rosa/ genus. (Row 39: /R. canina/ – „subject os our attention” --- but in the tittle is „Genus /Rosa/”, and much of the other literature also contains information on other species).
Although I believe that the first part of the manuscript should be significantly shortened, or changing the title and aim, my detailed opinion below is also aimed at improving this first part.
Only a small part of Figure 1 can be linked to the referenced text (Row 40).
Row 47: roses like cut or garden flowers – its a common information, why was chosen directly the Ref. [5] chosen like the source?
Row 65-66. The species…. sentence is not connected to this place in the text, the next sentence is more belong to the next paragraph, but connected together with the other text.
Subchapter 2: only the second half important for the current research topic. From the second part the seed properties are mainly important.
The photos in the manuscript are very nice, but not important from the view of germination, only the Figure 5 connected strong to the scientific topic.
Table 1: the main part of the data are from Ref. [32] – and marked with * (foot noted from USDA). Please use clear correct way of the citation source, original - open available USDA database which you can citate straight.
The subchapter 3.1 and 3.2 presents together the biochemical composition - no reason to divide it into two parts. If this part staying, it should put things together, avoiding repetition.
Row 156-157: the second part of the sentence please rephrase.
Subchapter 3.3: the subject of this subchapter has already been.
Row 200: the vary in colour and different varieties are not show non the Figure 5.
Row 210: Germination („seed” word is not necessary).
Rows 231-254: in my opinion inappropriate wording (in several places in the text): „some /R. canina/ seeds”, „many /R. canina/ seed” –where from we know how many, or which seed has the mentioned demand, and which has not?
Rows 242-261: mixed factors and manipulation processes
Row 263-273 – like introduction for the whole topic…
Rows 299-311 – it is also like introduction, not to the end of the subchapter…
Rows 318. Harvesting time/period – it is not a treatment!
Rows 325-332: temperature all over the storage? (or how long treatments?)
Row 365: concentration used?
Row 398: 13 years old publication – continuous research was published? should be researched in the future – this is opinion of authors of current manuscript? if yes, need be more highlighted.
Row 406 „might affect how well they germinated” – please rephase it.
Row 412-414. it is important fact, but not to this part.
Row 420-424: the genetic make-up of the seeds is important way (direction) of the research, but it need be relocated to different place in the text (logically not connected to here).
Row 440: compost activator – which?
Rows 451-471: here are also results connected to different temperatures!
Rows 481-490: if this text is still based on Ref [86], need be together in the same paragraph with the previous.
Rows 492-498 -already mentioned earlier.
Chapter 5: More information is repetition, already mentioned earlier.
Row 559-562: presumably the treatments and seed "population" were not the same in the two studies compared, so the results are not clearly comparable.
References: the list need be formatted in line with the requirements of the journal.
The authors have used very few references from the last some years (2017-), I consider it necessary to refresh the material.
IN SUMMARY, THE MANUSCRIPT IS RECOMMENDED FOR PUBLICATION ONLY AFTER A MAJOR REVISION AND ADDITIONS HAVE BEEN MADE.
THE FIRST HALF OF THE MANUSCRIPT (UP TO PAGE 9) NEEDS TO BE SIGNIFICANTLY SHORTENED (OR THE TITLE AND PURPOSE CHANGED), THE REST OF THE TEXT NEEDS TO BE RE-ORGANIZED, NEEDS TO BE STRUCTURED IN A LOGICAL ORDER, REPETITION SHOULD BE AVOIDED, AND RECENT RESEARCH FINDINGS SHOULD BE REVIEWED IN MORE DETAIL. IT IS THEN PROPOSED TO CONCLUDE BY IDENTIFYING THE APPROPRIATE RESEARCH DIRECTION(S).
AFTER REVISION, A NEW REVIEWER-PROCESS IS REQUIRED.
According to the title of the review, the authors summarize the scientific knowledge on seed germination in the genus Rosa.
Generally, although the manuscript gives a relatively good overview of the subject of the title, it contains a long "introduction" of less relevant knowledge (distribution, uses, botanical description of Rosa canina, etc.). This section, which can be called an introduction, goes all the way to line 183 on page 9, and since it contains few information related to germination, it should be shortened to a maximum of 20-30 lines. In this part, by the way, it is not quite clear (and sometimes highlighted in the later text) whether the aim was the examination really of /Rosa canina/ or the whole /Rosa/ genus. (Row 39: /R. canina/ – „subject os our attention” --- but in the tittle is „Genus /Rosa/”, and much of the other literature also contains information on other species).
Although I believe that the first part of the manuscript should be significantly shortened, or changing the title and aim, my detailed opinion below is also aimed at improving this first part.
Only a small part of Figure 1 can be linked to the referenced text (Row 40).
Row 47: roses like cut or garden flowers – its a common information, why was chosen directly the Ref. [5] chosen like the source?
Row 65-66. The species…. sentence is not connected to this place in the text, the next sentence is more belong to the next paragraph, but connected together with the other text.
Subchapter 2: only the second half important for the current research topic. From the second part the seed properties are mainly important.
The photos in the manuscript are very nice, but not important from the view of germination, only the Figure 5 connected strong to the scientific topic.
Table 1: the main part of the data are from Ref. [32] – and marked with * (foot noted from USDA). Please use clear correct way of the citation source, original - open available USDA database which you can citate straight.
The subchapter 3.1 and 3.2 presents together the biochemical composition - no reason to divide it into two parts. If this part staying, it should put things together, avoiding repetition.
Row 156-157: the second part of the sentence please rephrase.
Subchapter 3.3: the subject of this subchapter has already been.
Row 200: the vary in colour and different varieties are not show non the Figure 5.
Row 210: Germination („seed” word is not necessary).
Rows 231-254: in my opinion inappropriate wording (in several places in the text): „some /R. canina/ seeds”, „many /R. canina/ seed” –where from we know how many, or which seed has the mentioned demand, and which has not?
Rows 242-261: mixed factors and manipulation processes
Row 263-273 – like introduction for the whole topic…
Rows 299-311 – it is also like introduction, not to the end of the subchapter…
Rows 318. Harvesting time/period – it is not a treatment!
Rows 325-332: temperature all over the storage? (or how long treatments?)
Row 365: concentration used?
Row 398: 13 years old publication – continuous research was published? should be researched in the future – this is opinion of authors of current manuscript? if yes, need be more highlighted.
Row 406 „might affect how well they germinated” – please rephase it.
Row 412-414. it is important fact, but not to this part.
Row 420-424: the genetic make-up of the seeds is important way (direction) of the research, but it need be relocated to different place in the text (logically not connected to here).
Row 440: compost activator – which?
Rows 451-471: here are also results connected to different temperatures!
Rows 481-490: if this text is still based on Ref [86], need be together in the same paragraph with the previous.
Rows 492-498 -already mentioned earlier.
Chapter 5: More information is repetition, already mentioned earlier.
Row 559-562: presumably the treatments and seed "population" were not the same in the two studies compared, so the results are not clearly comparable.
References: the list need be formatted in line with the requirements of the journal.
The authors have used very few references from the last some years (2017-), I consider it necessary to refresh the material.
IN SUMMARY, THE MANUSCRIPT IS RECOMMENDED FOR PUBLICATION ONLY AFTER A MAJOR REVISION AND ADDITIONS HAVE BEEN MADE.
THE FIRST HALF OF THE MANUSCRIPT (UP TO PAGE 9) NEEDS TO BE SIGNIFICANTLY SHORTENED (OR THE TITLE AND PURPOSE CHANGED), THE REST OF THE TEXT NEEDS TO BE RE-ORGANIZED, NEEDS TO BE STRUCTURED IN A LOGICAL ORDER, REPETITION SHOULD BE AVOIDED, AND RECENT RESEARCH FINDINGS SHOULD BE REVIEWED IN MORE DETAIL. IT IS THEN PROPOSED TO CONCLUDE BY IDENTIFYING THE APPROPRIATE RESEARCH DIRECTION(S).
AFTER REVISION, A NEW REVIEWER-PROCESS IS REQUIRED.
Author Response
Comments and Suggestions for Authors
According to the title of the review, the authors summarize the scientific knowledge on seed germination in the genus Rosa.
Generally, although the manuscript gives a relatively good overview of the subject of the title, it contains a long "introduction" of less relevant knowledge (distribution, uses, botanical description of Rosa canina, etc.). This section, which can be called an introduction, goes all the way to line 183 on page 9, and since it contains few information related to germination, it should be shortened to a maximum of 20-30 lines. In this part, by the way, it is not quite clear (and sometimes highlighted in the later text) whether the aim was the examination really of /Rosa canina/ or the whole /Rosa/ genus. (Row 39: /R. canina/ – „subject os our attention” --- but in the tittle is „Genus /Rosa/”, and much of the other literature also contains information on other species). Response: first off all, many thanks for taking the time to help me improve this paper. The whole manuscript has been revised and restructured. It has now, a clearer line.
Although I believe that the first part of the manuscript should be significantly shortened, or changing the title and aim, my detailed opinion below is also aimed at improving this first part.
Response: these sections have been shortened. Thank you!
Only a small part of Figure 1 can be linked to the referenced text (Row 40). Response: with the new structure it has more sense.
Row 47: roses like cut or garden flowers – its a common information, why was chosen directly the Ref. [5] chosen like the source?
Row 65-66. The species…. sentence is not connected to this place in the text, the next sentence is more belong to the next paragraph, but connected together with the other text. Response: the paper has been arranged differently, it should be fine now.
Subchapter 2: only the second half important for the current research topic. From the second part the seed properties are mainly important.
The photos in the manuscript are very nice, but not important from the view of germination, only the Figure 5 connected strong to the scientific topic. Response: we consider that the figures give a better understanding to the topic for the readers. Anyway, all figures were revised. Thank you!
Table 1: the main part of the data are from Ref. [32] – and marked with * (foot noted from USDA). Please use clear correct way of the citation source, original - open available USDA database which you can citate straight.
The subchapter 3.1 and 3.2 presents together the biochemical composition - no reason to divide it into two parts. If this part staying, it should put things together, avoiding repetition. Response: the chapters have been changed, and the repetitions have been deleted.
Row 156-157: the second part of the sentence please rephrase. Response: it has been re-organized.
Subchapter 3.3: the subject of this subchapter has already been. Response: repetitions have been deleted.
Row 200: the vary in colour and different varieties are not show non the Figure 5. Response: unfortunately, this was the only original photo that the first author had. The first author is currently working on her Phd, on Rosa genus, more specifically, Rosa canina, hence the photo.
Row 210: Germination („seed” word is not necessary). Response: it has been corrected. Thank you!
Rows 231-254: in my opinion inappropriate wording (in several places in the text): „some /R. canina/ seeds”, „many /R. canina/ seed” –where from we know how many, or which seed has the mentioned demand, and which has not? Response: we have revised and corrected the entire manuscript.
Rows 242-261: mixed factors and manipulation processes. Response: we have revised and corrected the entire manuscript.
Row 263-273 – like introduction for the whole topic… Response: we have revised and corrected the entire manuscript.
Rows 299-311 – it is also like introduction, not to the end of the subchapter…Response: we have revised and corrected the entire manuscript.
Rows 318. Harvesting time/period – it is not a treatment! Response: we have revised and corrected the entire manuscript.
Rows 325-332: temperature all over the storage? (or how long treatments?) Response: we have revised and corrected the entire manuscript.
Row 365: concentration used? Response: we have revised and corrected the entire manuscript.
Row 398: 13 years old publication – continuous research was published? should be researched in the future – this is opinion of authors of current manuscript? if yes, need be more highlighted. Response: we have revised and corrected the entire manuscript.
Row 406 „might affect how well they germinated” – please rephase it. Response: we have revised and corrected the entire manuscript.
Row 412-414. it is important fact, but not to this part. Response: we have revised and corrected the entire manuscript.
Row 420-424: the genetic make-up of the seeds is important way (direction) of the research, but it need be relocated to different place in the text (logically not connected to here). Response: we have revised and corrected the entire manuscript.
Row 440: compost activator – which? Response: we have revised and corrected the entire manuscript.
Rows 451-471: here are also results connected to different temperatures! Response: we have revised and corrected the entire manuscript.
Rows 481-490: if this text is still based on Ref [86], need be together in the same paragraph with the previous. Response: we have revised and corrected the entire manuscript.
Rows 492-498 -already mentioned earlier. Response: the repetitions have been deleted. We have revised and corrected the entire manuscript.
Chapter 5: More information is repetition, already mentioned earlier. Response: we have revised and corrected the entire manuscript.
Row 559-562: presumably the treatments and seed "population" were not the same in the two studies compared, so the results are not clearly comparable. Response: we have revised and corrected the entire manuscript.
References: the list need be formatted in line with the requirements of the journal. Response: Done!
The authors have used very few references from the last some years (2017-), I consider it necessary to refresh the material. Response: we have revised and corrected the entire manuscript.
Dear reviewer,
Thank you very much for the insights, unfortunately, we were not able to properly respond to each of the points because, by the time we received this last review, the paper was already changed to the core. We restructured the entire paper, and reorganized the chapters. We have also deleted a great part of the manuscript, mainly in the first part, as you also suggested.
Thank you for all your suggestions, they were very constructive and useful. We used them in order to improve the final version of the manuscript.
We really hope that now the paper is in a ‘better shape’ and will be accepted for publication.
Thank you!